# Thrombosis in Paroxysmal Nocturnal Hemoglobinuria (PNH): From Pathogenesis to Treatment

**DOI:** 10.3390/ijms252212104

**Published:** 2024-11-11

**Authors:** Styliani Kokoris, Antri Polyviou, Paschalis Evangelidis, Elisavet Grouzi, Serena Valsami, Konstantinos Tragiannidis, Argyri Gialeraki, Dimitrios A. Tsakiris, Eleni Gavriilaki

**Affiliations:** 1Laboratory of Hematology and Blood Bank Unit, “Attikon” University General Hospital, Medical School, National and Kapodistrian University of Athens, 12462 Athens, Greece; stylianikok@gmail.com (S.K.); agialer@med.uoi.gr (A.G.); 2Department of Hematology and Lymphoma, BMT Unit, Evangelismos General Hospital, 10676 Athens, Greece; polyviouandri@gmail.com; 3Second Propedeutic Department of Internal Medicine, Aristotle University of Thessaloniki, 54642 Thessaloniki, Greece; pascevan@auth.gr (P.E.); konstantinos.tragiannidis@gmail.com (K.T.); 4Department of Transfusion Service and Clinical Hemostasis, “Saint Savvas” Oncology Hospital, 11522 Athens, Greece; grouzielisavet@gmail.com; 5Hematology Laboratory-Blood Bank, Aretaieion Hospital, National and Kapodistrian University of Athens, 11528 Athens, Greece; serenavalsami@yahoo.com; 6Department of Hemostasis and Thrombosis, University of Basel, 4001 Basel, Switzerland; dimitrios.tsakiris@unibas.ch

**Keywords:** bone marrow failure, complement, eculizumab, endothelium, paroxysmal nocturnal hemoglobinuria, ravulizumab, thrombosis

## Abstract

Paroxysmal Nocturnal Hemoglobinuria (PNH) constitutes a rare bone marrow failure syndrome characterized by hemolytic anemia, thrombotic events (TEs), and bone marrow aplasia of variable degrees. Thrombosis is one of the major clinical manifestations of the disease, affecting up to 40% of individuals with PNH. Venous thrombosis is more prevalent, affecting mainly unusual sites, such as intrabdominal and hepatic veins. TEs might be the first clinical manifestation of PNH. Complement activation, endothelial dysfunction, hemolysis, impaired bioavailability of nitric oxide, and activation of platelets and neutrophils are implicated in the pathogenesis of TEs in PNH patients. Moreover, a vicious cycle involving the coagulation cascade, complement system, and inflammation cytokines, such as interleukin-6, is established. Complement inhibitors, such as eculizumab and ravulizumab (C5 inhibitors), have revolutionized the care of patients with PNH. C5 inhibitors should be initiated in patients with PNH and thrombosis, while they constitute a great prophylactic measure for TEs in those individuals. Anticoagulants, such as warfarin and low-molecular-weight heparin, and, in selected cases, direct oral anticoagulants (DOACs) should be used in combination with C5 inhibitors in patients who develop TEs. Novel complement inhibitors are considered an alternative treatment option, especially for those who develop extravascular or breakthrough hemolysis when terminal inhibitors are administered.

## 1. Introduction

Paroxysmal nocturnal hemoglobinuria (PNH) constitutes a rare bone marrow failure disorder of hematopoietic stem cells caused by an acquired mutation in the phosphatidylinositol glycan class A gene (PIG-A), which leads to loss of glycosylphosphatidylinositol GPI-anchored proteins on the cell membrane [1]. As a result, the deficiency of complement regulatory proteins CD55 and CD59 on the surface of blood cells leads to complement-mediated formation of the membrane attack complex (MAC), causing intravascular hemolysis and platelet activation [2]. Identification of PNH cells is performed using flow cytometry, estimating the ratio of GPI-negative erythrocytes, neutrophils, and monocytes [3]. According to the International PNH Interest Group, PNH is classified into three subtypes: classic hemolytic PNH, PNH associated with bone marrow failure syndromes (e.g., aplastic anemia (AA) or myelodysplastic syndrome (MDS), and subclinical PNH) [4]. Interestingly, the presence of PNH clones in patients with bone marrow failure syndromes worsens the outcomes of allogeneic hematopoietic cell transplantation [5].

Complement inhibitors have revolutionized patient care in PNH since their Food and Drug Administration (FDA) approval [6]. Eculizumab (SOLIRIS^®^, Alexion Pharmaceuticals) and ravulizumab (ULTOMIRIS^®^, Alexion Pharmaceuticals) are designed to inhibit terminal complement activation at C5 [7]. Various other therapeutic C5 inhibitors (crovalimab, LFG316, REGN3918, ABP959, elizaria, and zilucoplan) are under ongoing clinical investigation for PNH treatment [8,9,10,11,12]. Currently, pegcetacoplan (Empaveli^®^, Apellis Pharmaceuticals), which inhibits proximal complement activation at C3, has been approved for the treatment of patients with PNH [13]. Moreover, improved hematological benefits have been observed in PNH patients treated with the Factor D inhibitor danicopan and the Factor B inhibitor iptacopan, both of which are inhibitors of the alternative complement system and were recently granted FDA approval. These new inhibitors often target proximal complement via innovative mechanisms and can be administered either subcutaneously or intravenously (IV) [14]. The role of complement activation and inhibition has been studied in various blood and non-hematological disorders [15,16,17,18,19].

Prevalent complications of PNH include thromboembolic events (TEs), smooth muscle dystonia, which might be manifested as dysphagia or erectile dysfunction, and chronic kidney disease [1,20]. The hypercoagulable state in PNH has been described by Luzzatto as “the most vicious acquired thrombophilic state known to medicine” [21]. Thrombosis represents one of the most common complications and the main cause of death in patients with PNH [22,23,24,25]. The factors contributing to the prothrombotic state observed in PNH include intravascular hemolysis, complement activation, and defects in fibrinolysis.

In this review, we intend to provide an up-to-date summary of the knowledge of thrombosis in PNH. A narrative review of the literature was performed using PubMed and Medline search engines to find the original articles. Full-text articles published in English, based on our search, were included, and all the available data were critically examined. The keywords of research were “Paroxysmal nocturnal hemoglobinuria” or “PNH” combined with “thrombosis”, “thromboembolic events”, or “anti-coagulation therapy”. We provide data about the incidence, risk factors, clinical characteristics, pathophysiology, and management of TEs affecting patients with PNH. In the era of novel and precise complement therapeutics, a better understanding of the prothrombotic state in PNH is crucial to delivering better care to those patients.

## 2. Incidence-Risk Factors

### 2.1. Incidence

PNH is a rare disorder, with an estimated incidence as high as 15.9 individuals per million worldwide [26]. PNH was first described by European physicians in the latter half of the 19th century, while several observational studies resulted in the establishment of PNH as a different clinical entity from march hemoglobinuria and paroxysmal cold hemoglobinuria [27]. TEs affect 37–39% of patients with classical PNH [22,24,28,29]. In a Greek cohort, it was reported that 39% of patients with PNH experienced at least one thromboembolic event [30]. The main cause of death in PNH before therapeutic complement inhibition was thromboembolism (TE). Thrombotic events in PNH patients were associated with high mortality and low survival rates [31]. The incidence of thrombosis after the broad use of eculizumab in clinical practice was reduced (relative reduction of thrombotic events 81.8%) [32,33].

### 2.2. Risk Factors

Patients with a large clone of CD59-negative polymorphonuclear cells (PMN) (PNH clones more than 50%) are at significant risk for thrombosis [24,34,35]. Data from the international PNH registry showed that 4.6% of patients with clone size <10%, 6.5% of those with a size between 10% and 50%, and 17.6% of PNH individuals with clone size larger than 50%, experienced TEs [25]. In this analysis, the patients were not treated with complement inhibitors at baseline. However, patients with PNH clones <50% might also experience an increased risk for thrombosis; thus, clinical education and vigilance are essential [36,37]. There is also a recent analysis of International PNH Registry data that suggests a correlation between clone size and disease burden/risk of major adverse vascular events (MAVEs), which could have predictive significance for doctors making decisions about how to treat PNH patients who are at risk of MAVEs [38]. MAVEs include TEs, amputation (nontraumatic, nondiabetic), myocardial infarction, transient ischemic attack, unstable angina, gangrene (nontraumatic, nondiabetic), and other MAVEs.

Other risk factors for TEs in PNH patients include lactate dehydrogenase (LDH) 1.5 or more × upper normal limit (ULN), previous thrombotic events, and the presence of clinical symptoms of PNH [28,36,37]. Furthermore, the presence of specific polymorphisms (rs495828, rs2519093) in the ABO gene has been associated with a probable increase in von Willebrand factor (VWF) and factor VIII levels, as well as an elevated thrombotic risk in those patients [39,40]. Elevated levels of anti-phospholipid antibodies were identified in a small study in eight PNH patients (61.5%), five of whom had a history of thrombosis. Dragoni et al. suggested that immune system involvement in PNH patients might play a role in the development of APA. However, these results were not corroborated by other studies [41]. The evaluation of inherited thrombophilic factors for TE in PNH has so far been only partially explored, with some relatively small studies showing no apparent increase in their frequency. Testing for these factors may help identify PNH patients at additional risk. However, its utility in predicting thrombosis recurrence in unselected patients with venous thrombosis is limited. The role of such testing in guiding treatment decisions in PNH remains unclear, and routine testing in all patients is not currently recommended [42]. The risk factors for TEs in PNH patients are summarized in Table 1.

### 2.3. Clinical Manifestations of TEs

TEs might be the first presentation sign in patients with PNH [43]. Venous thrombosis tends to be more common and may affect atypical sites such as hepatic (Budd–Chiari syndrome: BCS), mesenteric, renal, dermal, and cerebral veins [20,43]. Specifically, Devalet et al. reported that TEs are 85% of venous origin and 15% arterial, while 20.5% of patients experience thrombosis in more than one site at the same time [44,45]. Hill et al. showed that subclinical thrombosis is common in PNH patients, as assessed by high-resolution MR imaging [46].

BCS has been reported to affect 7.5–25% of PNH patients [47,48]. It has been suggested that all patients affected by BCS should be evaluated for the presence of PNH clones by flow cytometry [48]. In 19% of patients, visceral thrombosis precedes the diagnosis of PNH. Visceral vein thrombotic events also include mesenteric vein thrombosis, which presents as abdominal pain, bowel obstruction, possible rectal bleeding, and duodenal venous infraction [47,49]. An explanation is still required for the tendency toward intra-abdominal thrombosis in PNH. One theory suggests that CD59-negative neutrophils, which are commonly found in PNH, may be more prone to localize in liver microvessels. Here, they can readily interact with platelets, which are also more easily activated and release serine proteases, thereby concentrating procoagulant activity in this region. It is also quite interesting to investigate whether the PIG-A mutation affects endothelial cells (ECs) of splanchnic vessels [22].

Cerebral vein thrombosis, with thrombosis of the superior sagittal sinus being the most common, is prevalent in PNH patients, and it is associated with high mortality rates [50]. Cerebrovascular ischemia, as assessed by magnetic resonance imaging (MRI), affects patients with PNH, and it might be asymptomatic [51]. The central retinal vein or arterial thrombosis is a rare but potential complication in PNH. Thrombosis of skin veins and purpura fulminans might also be developed in PNH [52]. There have also been reports of hemorrhagic necrosis on the lip in patients with both PNH and COVID-19 infection [53]. COVID-19-induced endothelial damage is believed to contribute to the development of exaggerated hemorrhagic necrosis in these cases.

Deep vein thrombosis (DVT) is more common in individuals affected by PNH than in the general population [32,54]. Moreover, subclinical pulmonary embolism and pulmonary hypertension constitute a serious complication of PNH [55]. Thrombotic episodes of 41 patients with PNH from 14 different national hematology centers in Greece were recently analyzed: 16 patients (39%) experienced at least one episode of thrombosis, including 7 (43.8%) at diagnosis, 7 (43.8%) during the course of the disease and two (12.5%) patients prior to PNH diagnosis. In none of the 16 patients was the thrombotic event fatal. Nearly half of these individuals (n = 7, 43.8%) had multiple episodes of thrombosis during the course of their disease. The most common sites of thrombosis were intraabdominal veins. Three out of 26 patients developed thrombosis while on eculizumab [30].

Thrombosis of arterial origin might also be the first manifestation of PNH [22]. Arterial thrombosis is more frequent in PNH patients than in healthy subjects, and arterial and coronary arteries are the most commonly affected [29,56,57,58]. Although arterial thrombosis is considered a rare complication, studies have shown that this risk may be underestimated.

Patients who either are young or experience a thrombotic event in an uncommon site (e.g., visceral veins) have hemolytic anemia, or are affected by any cytopenia and experience an unexplained thrombotic event should be evaluated for the presence of PNH clone with flow cytometry [22].

Every patient with myeloproliferative neoplasm (MPN) who exhibits an unusual pattern of thrombosis, such as in the splanchnic veins or cerebral sinuses, should undergo testing for PNH [59]. It has been revealed that approximately 10% of MPN patients harbor a PNH clone, and this frequency increases to 17% when clones smaller than 1% are included [60]. The discovery of MPN driver mutations, particularly Janus Kinase 2 (JAK2), in GPI-deficient cells has sparked the hypothesis that these mutations could provide PNH cells with a natural growth advantage [61].

## 3. Pathogenesis of Thrombosis in PNH

### 3.1. Platelet Activation, Platelet-Derived Microparticles, and Platelet-Induced Complement System Activation

Complement-mediated cell injury leads to activation of PNH platelets, aggregation, and release of proinflammatory factors [22]. In PNH, platelet activation has been identified as a contributor to prothrombotic state and clot formulation [62,63,64]. Interestingly, normal platelet lifespan has been reported in PNH [65]. Furthermore, platelets seem to survive after MAC formation, while morphological adaptations after C5-b9 deposition on their surface have been described [66,67,68,69]. Those changes include depolarization of the membrane, loss of normal lipid asymmetry, secretion of alpha-granules, and fusion of granules with the cell membrane [44,66,70,71]. In addition, phosphatidylserine is manifested in the external leaflet of the activated platelet membrane [22]. Phosphatidylserine has a crucial role in the formation of prothrombinase and tenase complexes [72,73,74]. Thus, platelet-derived microparticles (PMPs) are released from the activated platelets [68]. Wiedmer et al. suggested that activated PNH platelets gave higher affinity for factor V binding compared to platelets from healthy controls [64]. Moreover, there is a crosstalk between activated platelets and neutrophils, leading to the expression of serine proteases and the activation of the blood coagulation cascade [75]. Hill et al. reported that platelet activation in PNH is implicated in other factors as well, such as impaired nitric oxide (NO) production, extravascular hemolysis, endothelial dysfunction, and reactive oxygen species (ROS) [22]. However, Grünewald et al. found diminished platelet reactivity and suggested that dysregulation in platelet functions in PNH might be implicated in the pathogenesis of thrombotic events in those patients [76]. PMPs, exhibiting prothrombinase activity as the activated platelets, express receptors for factor VIII and other coagulation cascade factors [73,74,77]. Nieuwland et al. showed that PMPs can in vivo induce the formation of thrombin [78]. Furthermore, PMPs express several molecules that contribute to the development of a prothrombotic state in PNH patients. These include glycoprotein Ib (GPIb), platelet endothelium adhesion molecule (PECAM-1), integrin glycoprotein IIb/IIIa (GpIIb-IIIa), P-selectin, and phosphatidylserine [44]. GPIb and GpIIb-IIIa are receptors of VWF, while P-selectin (as well expressed by activated platelets) activates the alternative pathway of the complement system, leading to a vicious cycle between thrombosis and inflammation [79,80,81,82]. When isolated PMPs are reintroduced into platelet-free pooled plasma without the addition of coagulation activators, they induce thrombin generation, indicating that microparticles generated in vivo can activate coagulation [78]. The classical pathway of the complement system can be activated by chondroitin sulfate deprived from activated platelets [82,83].

In a recent issue of Blood Advances, Mannes et al. discovered that hemolysis resulting from MAC formation induces prothrombotic activation of platelets and ECs by releasing the intracellular signal adenosine diphosphate (ADP) [84]. It also suggests that MAC formation, rather than upstream complement activation products, is necessary to induce thrombosis. When C3 or C5 complement components were blocked, there were no indicators of platelet activation. This work presents an interesting approach to connecting innate immunity, platelets, and thrombosis, offering a potential explanation as to why complement is a promising target in various complement-mediated diseases with significant thrombotic manifestations [85].

### 3.2. Intravascular Hemolysis, Toxicity of Free Hemoglobin, ROS Production, and Red Cell Pro-Thrombotic Microvesicles

Intravascular hemolysis is the major clinical feature in PNH and has been proposed to have a leading role in the pathogenesis of thrombotic events in PNH [86]. However, Griffin et al. suggested that significant intravascular hemolysis might not always be essential for thrombosis in PNH patients, and additional mechanisms seem to be implicated [87]. Clearance of free hemoglobin is performed by haptoglobin, hemopexin, and CD163-CD91 pathways [2,88]. Still, intravascular hemolysis can produce erythrocyte fragments and microparticles, which might promote activation of coagulation reactions and thrombin generation, as described below. Intravascular hemolysis leads to increased plasma concentration of free hemoglobin, which may have procoagulant properties [89,90]. In rats, platelet aggregation and adhesion in injured vessel walls were increased after the administration of purified human cross-linked hemoglobin (Hb)-ααHb [91]. Simionatto et al. showed that intravenous administration of hematin, an exogenous source of heme, in healthy subjects resulted in the development of peripheral thrombophlebitis [92]. The activity of metalloprotease ADAMTS13, cleaving protease of VWF, is reduced due to intravascular hemolysis [93,94]. During intravascular hemolysis, toxic species of iron are released, which is essential for the Fenton reaction (a process that involves the generation of hydroxyl radicals using ferrous iron and hydrogen peroxide) and the production of ROS [90,95]. PNH cells have been shown to be under oxidative stress, and reduced glutathione levels have also been reported [96]. Increased ROS production mediated by heme-induced NADPH oxidase (NOx), has been associated with loss of organization of membrane lipids, platelets activation, inflammation, cellular damage, and production of neutrophil extracellular traps (NETs), serving as a scaffold for the adhesion of platelets and red blood cells (RBCs) [97,98,99]. Extracellular DNA and histones found in NETs might contribute to thrombus formation, thrombin generation, and protein C inhibition [100,101]. Activation of Toll-like receptor 4 (TLR-4) in leukocytes promotes their rolling and adhesion to ECs.

Complement-mediated red cell damage leads to the production of red cell-derived microvesicles or membrane particles [102]. These microvesicles have been reported to have prothrombotic properties [102,103]. Erythrocyte membrane particles incorporate and accumulate heme within their membrane, facilitating the transfer of heme to ECs.However, in some studies, it was found that the levels of erythrocyte microvesicles were not statistically different between PNH patients and healthy subjects [62,68,104,105,106].

Beyond anemia and the loss of the anti-inflammatory functions of RBCs, hemolysis results in the release of damage-associated molecular patterns (DAMPs) such as cell-free hemoglobin (cfHb) and cell-free heme (cfHeme). These substances act through multiple receptors and signaling pathways, promoting a hyperinflammatory and hypercoagulable state [107]. The procoagulant phenotype associated with hemolytic disorders is a result of cfHb-induced endothelial dysfunction and platelet aggregation caused by direct interactions with cells, reduced NO bioavailability, and ROS generation. CfHeme induces a procoagulant state through multiple mechanisms. CfHeme, through intracellular pathways, leads to the activation of platelets. P-selectin allows the adhesion and activation of platelets. When CfHeme binds to monocytes, neutrophils, and endothelial cells, it stimulates these cells to produce tissue factor (TF), leading to increased coagulation activity [88].

### 3.3. Impaired NO Bioavailability

Increased levels of free hemoglobin due to intravascular hemolysis resulted in depletion of NO [89]. Hemolysis results in the release of arginase, an enzyme that disintegrates arginine, which is essential for NO synthesis [108,109]. Scavenging of NO leads to platelet activation and aggregation dysfunction, and local vasoconstriction, contributing to thrombosis [89,110,111,112]. Impaired production of NO has been associated with increased levels of P-selectin, leading to the activation of the complement system [81,112,113]. Shao et al. showed that diminished levels of NO in rats were associated with the formation of thrombus and increased fibrin production [114]. Eculizumab administration in PNH patients reduced the rate of NO depletion [115].

### 3.4. Endothelial Dysfunction—The Role of Leukocytes

Intravascular hemolysis has been associated with EC activation and TEs [116]. Free hemoglobin has prooxidant and proinflammatory effects, which result in EC dysfunction [110,117]. ECs are activated by heme in a TLR-4-dependent manner, which can lead to either the secretion of the contents of Weibel Palade bodies (WPBs), including VWF or to the generation of ROS that stimulate the increase in surface expression of adhesion proteins such as P-selectin and vascular cell adhesion molecule (VCAM-1) [95]. Moreover, the expression of P-selectin, E-selectin, intercellular adhesion molecule 1 (ICAM1), VCAM1, and VWF is induced [95,118]. As mentioned above, these molecules can activate complement and coagulation. Furthermore, the platelet GPIb receptor affinity for VWF is increased due to the presence of free hemoglobin [119].VCAM-1 has been shown to have prothrombotic properties [116]. Increased levels of endothelial-deprived microparticles (EMPs) have been reported in PNH patients, and their presence has been correlated with a higher risk of thrombosis [104,120,121,122]. EMPs are identified by the presence of endothelial cell markers such as CD144 and CD105, which indicates that PNH is characterized by chronic and persistent endothelial activation [22]. Eculizumab has been found to improve endothelial function markers and has been correlated with diminution of thrombotic risk [121]. Leukocytes interact with platelets and endothelial cells, forming a cellular bridge that can enhance thrombogenesis. The expression of adhesion molecules, such as P-selectin and E-selectin, is upregulated in response to complement activation and endothelial damage, facilitating the binding of leukocytes to the endothelium and promoting thrombus formation. In PNH, the underlying genetic mutation leads to a cascade of events at the transcription and protein levels, resulting in a complex interplay of cellular processes that favor thrombus formation. The interplay between platelet activation, coagulation cascade, and leukocyte adhesion is pivotal in the development of thrombosis.

### 3.5. Deficiency or Absence of Other GPI-Linked Proteins

#### 3.5.1. Defective Fibrinolysis

The urokinase plasminogen activator receptor (uPAR) is a GPI-linked protein expressed on neutrophils, ECs, and other cells like keratinocytes, fibroblasts, smooth muscle cells, megakaryocytes, and certain tumor cells [123]. After cleavage from the cell surface, soluble uPAR (suPAR) can be found in the blood and other organic fluids in all individuals, existing in three forms (I-III, II-III, and I) that have different properties related to their structural differences [124,125]. On the cell surface, pro-urokinase plasminogen activator(uPA)binds to uPAR via its growth factor domain, facilitating the conversion of pro-uPA to active uPA. Active uPA then cleaves the proenzyme plasminogen, producing active plasmin. Although uPAR is normally expressed at low levels, it is induced during leukocyte activation and differentiation in response to environmental stimuli such as smoking and certain RNA viruses [126]. Neutrophils contain intracellular reservoirs of uPAR, which are translocated to the plasma membrane upon activation.

Defective fibrinolysis has been described in PNH. Previous studies have shown that the membrane-bound form of uPAR is reduced or absent in granulocytes, while soluble uPAR levels are elevated in the patient’s plasma. Theoretically, suPAR might bind to plasma uPA, preventing its interaction with the active membrane-anchored form on residual normal cells. The combination of hemolysis and suPAR-related impairment in fibrinolysis in PNH may account for the higher incidence of thrombosis in patients whose disease is predominantly characterized by hemolysis. Multivariate analysis of factors associated with thrombosis in a cohort of 87 PNH patients revealed that LDH levels, male gender, and suPAR levels were independently linked to the time to the first thrombotic event [127].

In a publication by the International Study of Inflammation in COVID-19, a multinational observational study of patients hospitalized with COVID-19, researchers found that elevated suPAR levels were associated with an increased risk of thrombosis. The authors proposed that combining suPAR, a marker of the immune system, with D-dimer could enhance the accuracy of identifying which hospitalized COVID-19 patients are at high or low risk of blood clot formation [128]. Cell-associated plasminogen activation by PNH-affected neutrophils is severely impaired, and it has been proposed that this impairment may be causally related to the increased propensity for thrombosis in PNH [127]. Plasminogen activator inhibitor-1 is released by activated platelets and neutrophils in PNH and leads to fibrinolysis inhibition [129]. GPI-anchored co-receptor for TF pathway inhibitor (TFPI), which is an anticoagulant protein, is deficient in PNH, contributing to the development of thrombosis [130,131].

#### 3.5.2. Complement-Mediated Procoagulant Mechanisms

Complement activation plays a substantial role in the prothrombotic tendency observed in patients with PNH. The absence of C55 and CD59 complement-regulating proteins on PNH platelets leads to C5b-9-mediated expression of procoagulant enzyme complex prothrombinase [64,66]. C5a leads to the production of proinflammatory cytokines, such as interleukin-6 (IL-6) and tumor necrosis factor-a (TNF-A) [132,133]. IL-6 has been found to induce the expression of the IIa factor while suppressing the activity of ADAMTS13 [134,135]. Moreover, prothrombin, factor X, and factor XI can be activated by C5a [136,137]. Thrombin has been found to activate the complement system and induce the expression of C3 and C5 [138]. Thus, a chain reaction is established in which thrombosis activates complement, and in reverse, complement activation results in a prothrombotic state. Expression of TF, the primary initiator of the blood coagulation cascade, is induced by C5a signaling in neutrophils [139]. Furthermore, C5a may lead to both diminished protein S levels and protein C resistance, as suggested by Hill et al. [22]. Thrombosis in PNH at the time of infection can be attributed to pathogen-induced complement activation [22].

Interestingly, complement inhibition has been found to be the most effective treatment to prevent TEs in PNH individuals [32,140,141]. Eculizumab reduced the rate of TEs (82% relative reduction) in PNH patients, and it might be speculated that complement activation is one of the major mechanisms implicated in the prothrombotic state observed in those patients [32]. So, the complement and coagulation cascades are interconnected and work together during inflammation and hemostasis, which may ultimately lead to pathological thrombosis. The mechanisms implicated in the pathogenesis of thrombosis in PNH are summarized in Figure 1.

## 4. Management of Thrombosis in PNH

### 4.1. Acute Treatment of Thrombosis

Thrombotic events in PNH patients constitute a hematological emergency. The first therapeutic intervention to be made is the administration of unfractionated heparin or low-molecular-weight heparin in the absence of major contraindications (with dose adjustment in patients with impaired renal function). C5 therapeutic inhibitor (eculizumab or ravulizumab) should be initiated as soon as possible in combination with anticoagulant treatment in patients who have never received a complement inhibitor before and experience an acute thromboembolic event. Anticoagulation alone is ineffective in preventing the extension of the recurrence of complement-mediated TE [8,142,143,144]. Eculizumab 600 mg is administered IV once a week during the first phase of treatment (four weeks), followed by 900 mg for the fifth dose 1 week later. Then, 900 mg of eculizumab should be given IV every two weeks after the induction therapy. In clinical trials, ravulizumab was not found to be less effective than eculizumab. The major advantage of ravulizumab (longer acting than eculizumab) is that after the introduction of therapy (two weeks), it can be administered with an IV every eight weeks. The rapid onset of terminal complement blockade is crucial. Therefore, we avoid using subcutaneous therapies, as they are gradually absorbed and can take longer to achieve therapeutic levels [145]. In patients with underlying bone marrow failure, the management is similar to those with classical PNH when symptoms of hemolysis are prominent.

In patients already taking eculizumab, there is a risk of developing thrombosis, for example, during periods of breakthrough hemolysis (BTH) caused by infection. Patients need then to immediately take an additional dosage of eculizumab (a 300 mg dose increase is suggested) and therapeutic anticoagulation. CH50 can be a valuable tool for evaluating complement blockade, especially when there are concerns about BTH. LDH also is a very suitable biomarker of hemolysis [9,143]. It is known that intravascular hemolysis is characterized by an elevated reticulocyte count and high serum LDH levels, along with low serum haptoglobin, in the absence of hepatosplenomegaly. Hemosiderin also can be found in urine sediment and may accumulate in the kidneys. Laboratory monitoring of patients with PNH is important for evaluating their response to treatment. Only limited data on biomarkers in patients treated with proximal inhibitors have been published; however, more data and new biomarkers are anticipated as the development of these inhibitors progresses [146].

If a patient is already taking warfarin for therapeutic anticoagulation, a reassessment of international normalized ratio (INR) values (or levels of anti-Xa in patients using low molecular weight heparin, LMWH) should be used to determine effectiveness and adherence. Dose adjustment or alternative anticoagulation is recommended if therapeutic levels are not achieved (such as warfarin switch to LMWH). Monitoring the progression of any additional underlying illnesses is also crucial for patients receiving C5 inhibitors and experiencing thrombosis. In a Greek cohort, it was reported that out of 26 patients treated with eculizumab, three were documented to have thrombotic events. Each of these three patients had a concurrent underlying disease, such as primary myelofibrosis (PMF), polycythemia verra (PV), and homozygous thalassemia [30].

Failure to respond to eculizumab has been documented in a subset of Asian PNH patients [147]. These patients had a genetic variant of C5, specifically a missense mutation resulting in c.2654G>A (p.Arg885His). It is advisable to consider C5 genotyping for such individuals. Additionally, there is awareness of the hypomorphic variant of CR1, which is linked to increased surface C3 opsonization and a poor response to eculizumab due to extravascular hemolysis. In such cases, it is recommended to explore alternative treatments with novel complement inhibitors. Switch-over from C5 to C3 inhibitor therapy also may be considered in case of unprovoked thromboembolic events during C5 inhibitor therapy [148]. In Figure 2, an algorithm for the use of complement inhibitors in PNH patients is suggested.

In patients without access to eculizumab, lifelong anti-coagulation therapy has been recommended after a thromboembolic event [22,32]. Continuing anticoagulation with vitamin K antagonists is generally recommended in the long term if there are no contraindications. Otherwise, Brodsky suggested an overlap of anticoagulation and complement inhibition for 3 to 6 months. Also, he suggested that the administration of anticoagulation therapy should be terminated once the symptoms of thrombosis are resolved and the patient is on a complement inhibition and in a steady state (LDH < 1.5 × the upper normal limit) [143,149]. The absence of information regarding the risks versus benefits of this recommendation necessitates a discussion with the patient and critical thinking about it. Normally, in non-PNH patients, a thrombotic event has a relative risk for relapse of 14% for the first year, diminishing every year [150]. After 3–4 years, this risk comes to the lowest point of about 1–4% per year, which equals the bleeding risk of anticoagulation. Otherwise, the risk of bleeding and thrombocytopenia complications is decreased when anticoagulation is stopped. Prolonged anticoagulation may be considered for patients with additional risk factors for thrombosis or a history of severe thrombotic events. Patients with BCS should receive anticoagulant therapy as soon as possible for an indefinite period of time. Questions have been raised about how long long-term anticoagulation should be continued and whether patients on long-term anticoagulation require a dose reduction of the anticoagulant. These questions highlight the need for individualized treatment approaches supported by ongoing research and clinical trials to provide clearer guidance on long-term anticoagulation management.

For patients on eculizumab with a prior history of thrombosis, it is advisable to recommend the continuation of anticoagulation unless there are contraindications or clear evidence supporting the discontinuation of anticoagulation [32]. Selecting the appropriate anticoagulant dosage requires careful consideration of the patient’s comorbidities, which may include low platelet count (PLT), chronic liver disease, and moderate to severe renal failure [151]. For patients with thrombocytopenia and PLT between 149,000 and 50,000/μL LMWH, vitamin K antagonists and direct oral anticoagulants (DOACs) in full therapeutic dosage can be used, in those with PLT between 50,000 and 25,000/μL LMWH in 50% of the therapeutic or prophylactic dose is recommended, while when PLT < 25,000/μL anticoagulation is contraindicated. In patients with kidney failure and glomerular filtration rate (GFR) of 30–50 mL/min LMWH, vitamin K antagonists and DOACs in full therapeutic dosage are considered safe. However, in those patients, dabigatran and edoxaban need dose reduction. Moreover, when GFR is between 15–30 mL/min, LMWH in 50% of the dose, vitamin K antagonists, and some DOACs (apixaban, edoxaban, and rivaroxaban) with dose reduction are recommended. In patients with GFR below 15 mL/min, only vitamin K antagonists are recommended. When vitamin K antagonists are used, the INR target range is 2.0 to 3.0. However, more experience and extended follow-up are needed before recommending the discontinuation of anticoagulation in all PNH patients.

Patients receiving eculizumab should avoid plasma products unless it is an emergency, as these products contain elevated complement levels, leading to a higher risk of thrombosis and hemolysis due to the loss of complement blockade. In the event of an emergency requiring plasma products, an immediate additional dose of eculizumab is necessary. PNH is also characterized by platelet activation, which might induce the release of platelet factor 4 (PF4). As a result, exposure to heparin may make these patients more likely to develop heparin/PF4-dependent antibodies [152]. Heparin-induced thrombocytopenia has been reported to occur in some cases, so fondaparinux is an alternative anticoagulant. New oral direct thrombin and factor Xa inhibitors could be useful in these cases. Heparin also has the potential to make thrombosis worse in some circumstances by triggering the complement system. Cyclooxygenase system inhibitors like aspirin, ibuprofen, or sulfinpyrazone could serve as alternatives.

For refractory and life-threatening cases of thrombosis, thrombolytic therapy with tissue plasminogen activator (tPA) (tPA is administered as four consecutive intravenous infusions of 0.25 mg/kg over 6 h) has been used [153]. The BDS, as mentioned above, is a typical manifestation of thrombosis in PNH patients and might affect both the small and the large hepatic veins. Immediate commencement of eculizumab is recommended (can reduce mortality and long-term sequelae) along with anticoagulant treatment. Angioplasty of the inferior vena cava or hepatic veins, or transjugular intrahepatic portosystemic shunt (TIPS) may be required. In cases of splenomegaly secondary to portal or splenic vein thrombosis with concomitant hypersplenism, a procedure known as selective embolization of the splenic artery is performed. This procedure aims to reduce the size of the spleen and improve cytopenia.

DOACs have not been widely studied as antithrombotic treatment for PNH patients experiencing acute-phase thrombosis. Nevertheless, some centers routinely administer DOACs for at least 3–6 months and continue until intravascular hemolysis is controlled after initiating anti-complement therapy [154]. In a recent study by Gurnari and co-authors, a large real-world cohort of patients with PNH from four US centers to explore features, predictors of TE, and anticoagulation strategies were accrued [155]. Among 267 patients followed up for a total of 2043 patient-years, 56 (21%) developed TEs. These occurred at disease onset in 43% of cases, involving more frequently the venous system, typically as BDS. The rate of TEs was halved in patients receiving complement inhibitors (21 vs. 40 TEs per 1000 patient-years in untreated cases, with a 2-year cumulative incidence of thrombosis of 3.9% vs. 18.3%, respectively) and varied according to PNH granulocytes and erythrocytes clone size, type, disease activity parameters, as well as number (≥2 mutations, or less) and variant allelic frequency of PIGA mutations. Anticoagulation with warfarin (39%), DOACs (37%), and LMHW (16%) were administered for a median of 29 months [interquartile range (IQR) 9–61.8]. No thrombotic recurrence was observed in 19 patients treated with DOACs at a median observation of 17.1 months (IQR, 8.9–45), whereas 14 cases discontinued anticoagulation without TE recurrence at a median time of 51.4 months (IQR, 29.9–86.8).

### 4.2. Effects of the Complement Inhibitors on TE in Patients with PNH

In the 2007 report by Hillmen and co-authors, the effect of long-term treatment with eculizumab on the prespecified clinical outcome of TEs was evaluated on an intention-to-treat basis in a multinational phase 3 open-label extension study, which enrolled patients from three independent eculizumab PNH clinical studies [32]. In 195 patients who received the C5 inhibitor eculizumab in clinical trials between 2002 and 2005, thrombosis rates decreased from 7.37 events per 100 patient-years (1683 patient-years of exposure) without complement inhibition (i.e., before eculizumab treatment) to 1.07 events per 100 patient-years (281 patient-years of exposure) with eculizumab. There was an 82% reduction in the TE rate.

The complement component 5 (C5) inhibitor ravulizumab demonstrated non-inferiority to eculizumab following 26 weeks of treatment in complement inhibitor-naïve and complement inhibitor-experienced patients with PNH in the study of Kulasekhara and co-authors [156]. The non-inferiority of ravulizumab compared to eculizumab is demonstrated across several endpoints, including the normalization of LDH values, transfusion avoidance, percentage of patients with stabilized hemoglobin levels, BTH, and quality of life. Symptoms such as fatigue, hemoglobinuria, abdominal pain, dyspnea, and MAVEs, including TEs, were also assessed. The thrombosis rate with C5 inhibitor ravulizumab treatment was 1.21 events per 100 patient-years among 434 patients in the extension period (662 patient-years of exposure).

In a recent report of 509 patients with PNH receiving C5 inhibitors (eculizumab and/or ravulizumab) from May 2002 to July 2022 in the United Kingdom, 23 patients had a thrombotic event, consistent with a thrombosis rate of 0.73 events per 100 patient-years eculizumab and ravulizumab are safe and effective therapies that reduce mortality and morbidity in PNH [157]. However, further efforts are needed to reduce mortality in patients with concomitant bone marrow failure. 4–27% of PNH patients on therapeutic C5 inhibition experience hemolytic extravascular anemia due to C3b activation [158]. This might be explained by the fact that thrombin generation might activate C3b and/or residual C5a [159]. Since RBCs are no longer rapidly destroyed by the membrane attack complex, C3 fragment deposition can occur on the cell membrane, potentially leading to opsonization and subsequent extravascular clearance [160]. The recently approved C3 inhibitor pegcetacoplan, designed to block extravascular hemolysis, has been shown to improve anemia and hemolytic laboratory markers to a greater extent than eculizumab in patients with PNH [13]. Pegcetacoplan administration (subcutaneous injection) is considered an option for patients with inadequate response to eculizumab or ravulizumab.

In a recent analysis, all patients who received ≥1 dose(s) of pegcetacoplan in 7 clinical trials and in the post-marketing setting in the United States, Europe, and the rest of the world as of 13 November 2022 were included [161]. Four hundred and sixty-four patients with PNH had accumulated 619.4 patient-years of pegcetacoplan exposure across completed and ongoing clinical trials, as well as in the post-marketing setting. Seven TEs were reported: five in clinical trials (two in the same patient) and two in the post-marketing setting. The overall thrombosis rate was 1.13 events per 100 patient-years (1.22 events per 100 patient-years in 409.4 years for clinical trials and 0.95 events per 100 patient-years in 210.0 years for the post-marketing setting). It is also suggested that if a patient’s clinical condition deteriorates in the setting of BTH, for example, due to thrombosis or renal failure despite pegcetacoplan dosing intervention (a single IV dose of 1080 mg or 1080 mg subcutaneous on three consecutive days according to investigators’ clinical judgment), a single dose of eculizumab should be administered, if possible, to reduce the risk of life-threatening complications [162].

A 307 open-label extension study assessed the long-term safety and efficacy of pegcetacoplan in patients with PNH from five clinical trials [PHAROAH and PADDOCK (phase 1), PALOMINO (phase 2), and PEGASUS and PRINCE (phase 3)] [163]. Pegcetacoplan maintained the normalization of hematologic parameters, including hemoglobin concentrations, LDH concentrations, and FACIT-Fatigue scores. The adverse event profile for pegcetacoplan was consistent with that established in pivotal trials, and there were no reports of thrombotic events or meningococcal infections in the open-label at the data cutoff.

Despite current treatments, patients may still experience BTH, which can be recognized by signs and symptoms of intravascular hemolysis, such as hemoglobinuria, a significant rise in serum LDH, and a rapid drop in hemoglobin levels. Various definitions and reporting criteria for BTH have been used in complement inhibitor trials, and no standardized definition has been established. The causes of BTH can be primarily pharmacokinetic, resulting from low or insufficient drug levels that lead to suboptimal complement inhibition, or pharmacodynamic, where infections or other inflammatory conditions trigger (Complement-amplifying conditions or CACs) robust complement activation that temporarily overrides the drug-induced C5 blockade. Questions arise regarding how to minimize the risk of thrombosis in the case of BTH in situations such as vaccination, planned or unplanned surgery, trauma, or other CACs, as well as during pregnancy and the postpartum period. There is an option to begin prophylactic anticoagulation until hemolysis resolves, assuming the patient is not already on anticoagulation therapy. This strategy aims to lower the risk of thrombosis during episodes of active hemolysis [164].

BTH is one of the most concerning adverse events associated with complement inhibitors, and it may be more severe with proximal inhibitors compared to anti-C5 therapies [165]. With C5 inhibition (e.g., eculizumab), if the inhibition is incomplete, only one MAC is formed for each C5 molecule that escapes inhibition. In contrast, with incomplete pathway inhibition on pegcetacoplan, each C5 convertase enzyme can catalyze the cleavage of multiple C5 molecules, leading to the formation of several MACs. This increases the potential for significant BTH [156].

Combination treatment with a C5 and a C3 inhibitor may be required for a number of patients, as shown by this real-world data and as proposed by Notaro and Luzzatto [166]. Danicopan targets Factor D, a cofactor essential for C3 opsonization and alternative pathway activation and was found to be effective as adjuvant therapy. The combined approach could prevent C3 binding to PNH red cells, thereby reducing extravascular hemolysis while also helping to prevent massive BTH [159,165,167,168].

The PNH classification proposed by the International PNH Interest Group is based on the presence of clinical symptoms and laboratory evidence of hemolysis or thrombosis, signs of bone marrow failure such as AA and MDS, as well as cytometric analysis of the clone size of cells affected by the GPI defect. Individuals with a GPI(-) cell clone below 1% are considered healthy or non-PNH, but those who test positive are at risk of progressing to PNH.

A diagnosis of PNH does not automatically warrant the initiation of C5 inhibitor treatment. Eculizumab therapy should be started in patients with severe hemolysis, marked by an increasing LDH concentration (≥1.5 times the upper limit of normal [ULN]) and the presence of symptoms or conditions such as anemia with a hemoglobin level below 7 g/dL or below 10 g/dL in patients with cardiovascular symptoms; PNH-related thrombosis; complications of hemolysis, including worsening renal failure and pulmonary hypertension manifested by dyspnea; abdominal pain, dysphagia, or erectile dysfunction; and pregnancy, particularly in cases with previous miscarriages or other pregnancy complications [169]. In the population of patients with unexplained thrombosis, a PNH clone may be the cause.

According to a recent paper by Goh and colleagues, complement C5 inhibitors are indicated for the treatment of patients with PNH, with increased hemolysis (LDH > 1.5 ULN), granulocyte PNH clone >10%, and one or more of the following criteria:Clinical symptoms indicative of high disease activity (weakness, fatigue, hemoglobinuria, abdominal pain, dyspnea, anemia with Hb < 10 g/dL), thrombosis, dysphagia, and/or erectile dysfunction), regardless of transfusion history;History of TEs requiring anticoagulant therapy due to PNH;History of regular transfusions (at least four packs of RBCs over the past 12 months) due to hemolysis;Organ damage due to hemolysis (chronic renal failure or repeated episodes of acute renal failure; chest pain with New York Heart Association class III or IV; respiratory failure or an established diagnosis of pulmonary hypertension; and/or smooth muscle dystonia);Pregnancy with a high risk of thrombosis or a history of gestational complications [148].

The ongoing International PNH Registry (NC1374360) is the largest prospective, global, observational study of patients with PNH to date, continuously collecting patient data since its initiation in 2007. A deeper understanding of the prognostic value of clone size may better inform clinical decision-making and improve patient outcomes [38].

## 5. Vaccination in PNH

It is strongly advised that all patients on therapeutic C5 inhibitors should have a tetravalent vaccination against Neisseria meningitides serotypes ACYW135 and serogroup B recommended for all age groups and should be administered at least two weeks before receiving the first dose of eculizumab [170]. In order to prevent meningitis, in patients who need to receive eculizumab urgently, preventive antibiotics like ciprofloxacin may be given concurrently and continued for 14 to 21 days following their immunization. Vaccination is occasionally insufficient to prevent meningococcal infection, so all patients on therapeutic C5 inhibitors should receive penicillin, although the benefits and risks have not yet been established. Alternatively, a macrolide like azithromycin might be administered if there is a penicillin allergy. It is required that patients with suspected meningococcal infections be instructed to begin empirical treatment with 750 mg of ciprofloxacin (“pill in pocket” approach).

Revaccination is usually recommended every 2.5–3 years. Protection against meningococcal promptly after eculizumab treatment. Vaccines against Hemophilus influenzae type B and pneumococcus should also be administered.

There is a recent presentation of a rare case of PNH developing following a COVID-19 infection, a scenario that has been infrequently reported in the literature. Treatment with ravulizumab during COVID-19 pneumonia was deemed safe and cost-effective and resulted in a favorable clinical outcome in a patient with PNH. Complement inhibitors may potentially be effective in managing SARS-CoV-2-mediated complement activation-dependent hemolysis in PNH patients [171,172].

## 6. Prevention of Thrombosis

The efficacy of warfarin as a primary prophylaxis for the reduction of TEs in PNH is debatable, and it has been suggested that it might be insufficient [35,54,153]. In the study of Hall et al., patients with large PNH clones (PNH granulocyte clone size > 50%) and no contraindication to anticoagulation (such as low PLT that is observed in many PNH patients) were offered warfarin prophylaxis despite the risk of hemorrhagic events [29]. Moreover, TEs on patients who received primary anticoagulant prophylaxis are not uncommon [22]. Brodsky suggests that in patients who do not meet the criteria for eculizumab therapy, prophylactic anticoagulation should not be initiated. Possible exceptions may include patients with persistently elevated d-dimer levels (cut-off value in healthy individuals < 500 ng/mL), pregnant PNH patients, and patients in the perioperative period [143]. Using data from the International PNH Registry, a larger clone size (>30% clone size)at the baseline seems to indicate a higher disease burden and an increased risk of TEs and MAVEs. This information may guide physicians in making decisions when managing PNH patients at risk of experiencing TEs or other MAVEs [38].

Moreover, a multivariate analysis from the South Korean National PNH Registry showed that PNH patients with elevated hemolysis (LDH levels ≥ 1.5 times the ULN) at diagnosis were at significantly higher risk for TE than patients with LDH < 1.5 × ULN. The combination of LDH ≥ 1.5 × ULN with the clinical symptoms of abdominal pain, chest pain, dyspnea, or hemoglobinuria was associated with a greater increased risk for TE than elevated hemolysis or clinical symptoms alone [36].

Primary prophylaxis is not suggested for patients who receive a C5 complement inhibitor [8]. According to the Consensus statement for diagnosis and treatment of PNH in patients without eculizumab and with high PNH clone size (granulocyte clone > 50%), high level of D dimer, pregnancy, perioperative condition and other associated thrombophilic risk factors, including heritable thrombophilia, without known contraindication to anticoagulation and PLT stable (>100 × 10^9^/L), primary prophylaxis should be given. Secondary prevention with eculizumab is appropriate in patients who have already had a thromboembolic event related to PNH. Consider a DOAC for secondary VTE prevention only if the patient’s disease is well-controlled on eculizumab. DOACs may effectively prevent venous TE but increase the risk of bleeding complications [173].

## 7. Treatment of Thrombosis in Special Patients Groups

### 7.1. Pregnancy

Management of PNH during pregnancy is demanding, and collaboration between hematologists and gynecologists is crucial [143]. Interestingly, complement system activation is increased during pregnancy [174,175,176]. Thrombosis is associated with significant risks for both the mother and the fetus (miscarriages or premature births). According to studies, the maternal death rate in PNH patients during pregnancy and the early postpartum period is estimated to be between 12% and 21%. Given the fact that eculizumab does not cross the placenta, eculizumab is the C5 inhibitor used in pregnant women with PNH. Kelly et al. performed an analysis of data from the international PNH registry on 61 pregnant women with PNH and found no maternal deaths and three fetal deaths (4%) in the study group. In 23 pregnancies, either the eculizumab dose was increased, or the dosing interval was decreased. Moreover, they suggested that eculizumab was not present in breast milk samples [177]. Eculizumab treatment may be continued in pregnant women with PNH, especially those with associated risk factors for thrombosis. The dose of eculizumab may be increased in the third trimester or in cases of BTH, depending on individual circumstances. Treatment with eculizumab may be continued for at least 6 weeks postpartum [148]. LMWH as an anticoagulation therapy is suggested during pregnancy and post-partum (for 3 months) in PNH-pregnant women [142,143]. There are not enough data about the ravulizumab use during pregnancy [143]. Studies on ravulizumab efficacy and safety, as well as TE prophylaxis during pregnancy, are essential for this patient group. Breastfeeding also should not commence during therapy with ravulizumab and should be discontinued for eight months after due to the possibility of significant adverse effects in a nursing child.

### 7.2. Pediatric Patients

PNH is an ultra-rare cause of thrombosis in pediatric patients [178]. In the observational study of Curran et al., 12 consecutive PNH patients were enrolled, and thrombosis developed in 6 of them [179]. TEs occurred in unusual sites, such as portal veins, hepatic veins, and cerebral sinuses. Treatment with C5 complement inhibitors (eculizumab, ravulizumab) constitutes also the main option in these patients. In the recently published prospective study of Chonat et al., ravulizumab was found to be a safe and effective option for children with PNH in both eculizumab naïve or experienced patients [180]. More real-world studies examining the role of complement inhibitors in children with PNH are essential. However, this might be a challenging task, given the rarity of PNH in pediatric settings. It has to be highlighted that children at less than 2 years of age, or those without vaccination against *N. meningitidis*, should get vaccinated before the initiation of treatment and receive prophylactic antibiotics post-vaccination for the appropriate time course [169].

## 8. Conclusions

Thrombosis constitutes the most life-threatening complication of PNH. TEs might be the first manifestation of PNH. In the era of therapeutic complement inhibition, however, the prevalence of TEs has significantly been diminished. Complement activation, intravascular hemolysis, and endothelial dysfunction have a crucial role in the development of the prothrombotic state observed in PNH patients. However, we have to mention that other established prothrombotic risk factors might also be implicated in the pathogenesis of TE. Additional research is still required to delve into thrombosis-related factors among patients with PNH. Management and prevention of thrombosis are still debatable. However, eculizumab and ravulizumab transformed the care of patients with PNH. Future studies should focus on the efficacy of novel proximal complement therapeutics on thrombotic risk reduction of those patients and the effectiveness of other anticoagulant drugs (such as DOACs) in the management of thrombosis in PNH patients. Also, a combined-target (Dual inhibition) approach to treating PNH is indeed promising, and the current data indicate that it has the potential to be effective. Real-world data and multicenter collaboration are crucial to achieve better outcomes for our patients.

## Figures and Tables

**Figure 1 ijms-25-12104-f001:**
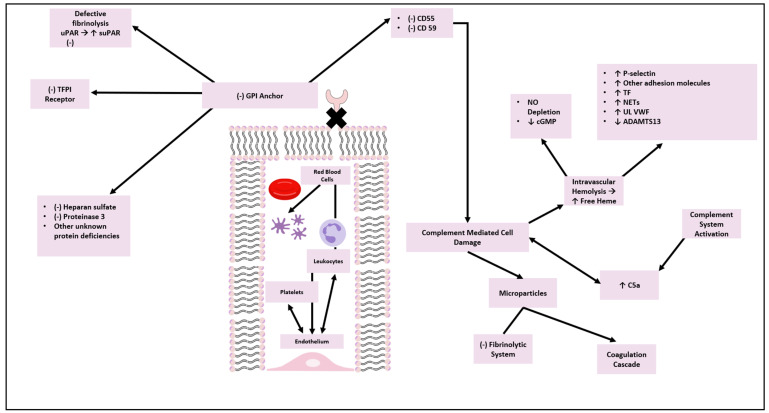
Pathogenesis of thrombosis in PNH. uPAR: urokinase plasminogen activator receptor, suPAR: soluble urokinase plasminogen activator receptor, TFPI: tissue factor pathway inhibitor, GPI: glycoprotein, NO: nitric oxide, cGMP: cyclic guanosine monophosphate, TF: tissue factor, NETs: neutrophil extracellular traps, UL VWF: ultra-large von Willebrand factor, ADAMTS13: on Willebrand factor-cleaving protease.

**Figure 2 ijms-25-12104-f002:**
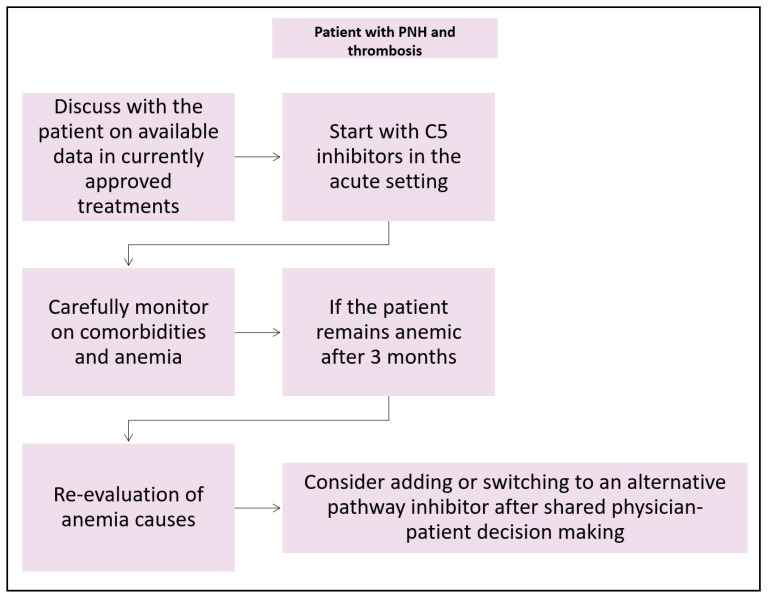
Suggested use of complement inhibitors in PNH patients. PNH: paroxysmal nocturnal hemoglobinuria.

**Table 1 ijms-25-12104-t001:** Risk factors for thrombotic events in PNH.

Risk Factors	References
Large clones of CD59-negative polymorphonuclear cells (PNH clones more than 50%)	[24,32,34]
Lactate dehydrogenase 1.5 or more × upper normal limit	[26,34,35]
Previous thrombotic events	[26,34,35]
Presence of clinical symptoms of PNH	[26,34,35]
Polymorphisms (rs495828, rs2519093) in the ABO gene	[37,38]
Elevated levels of anti-phospholipid antibodies ^1^	[39]

^1^ Data from a study with a relatively small number of study participants. PNH: Paroxysmal nocturnal hemoglobinuria.

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
