# Peer review of "Thrombosis in Paroxysmal Nocturnal Hemoglobinuria (PNH): From Pathogenesis to Treatment"

_ijms, 2024, doi:10.3390/ijms252212104_

Round 1
Reviewer 1 Report
Comments and Suggestions for Authors
This is a good manuscript that should be published.
A few points could be enhanced to make it even better and reader-friendly.
A section on clinical features and treatment of PNH in children would be helpful and would make a paper more universal.
Small susections with section title to break and structure the text would be advisable.
A decision tree /schematic representation of factor influencing decsions and treatment, perhaps in the 'How I treat' format would be very halpful for the readers and would make the paper more attractive.
Some gathering of major ppoints would be usueful, at the moment the paper is a bit of encyclopedia.
Author Response
Reviewer 1
This is a good manuscript that should be published.
Reply: We are thankful for the positive feedback. Furthermore, we would like to thank the reviewer for the time dedicated on reading and reviewing our work.
A few points could be enhanced to make it even better and reader-friendly.
A section on clinical features and treatment of PNH in children would be helpful and would make a paper more universal.
Reply: What an interesting idea! We added a section regarding the treatment approach of PNH in pediatric patients (7.2 Pediatric patients).
“PNH is an ultra-rare cause of thrombosis in pediatric patients[177]. In the observational study of Curran et al., 12 consecutive PNH patients were enrolled, and thrombosis developed in 6 of them [178]. TEs occurred in unusual sites, such as portal vein, hepatic veins, and cerebral sinuses. Treatment with C5 complement inhibitors (Eculizumab, Ravulizumab) constitutes also the main option in these patients. In the recently published prospective study of Chonat et al., Ravulizumab was found to be a safe and effective option for children with PNH in both Eculizumab naïve or experienced patients[179]. More real-world studies, examining the role of complement inhibitors in children with PNH are essential. However, this might be a challenging task, given the rarity of PNH in pediatric settings. It has to be highlighted that children at less than 2 years of age, or those without vaccination against N. meningitidis, should get vaccinated before the initiation of treatment and receive prophylactic antibiotics post-vaccination for the appropriate time course [180].” (Lines 724-736)
Small susections with section title to break and structure the text would be advisable.
Reply: Thanks for this comment. We made the essential changes throughout the manuscript as you suggested.
A decision tree /schematic representation of factor influencing decsions and treatment, perhaps in the 'How I treat' format would be very halpful for the readers and would make the paper more attractive.
Reply: We are thankful for this suggestion. We added a scheme (Figure 2, Page 11) regarding the use of complement inhibitors in PNH patients with thrombosis naive to complement inhibition.
Some gathering of major ppoints would be usueful, at the moment the paper is a bit of encyclopedia.
Reply: Thanks for this interesting idea. We enhanced the conclusions section, highlighting the points of our review.
Reviewer 2 Report
Comments and Suggestions for Authors
Content suggestions:
1. Can the Authors specify the epidemiology and history of the PNH ?
2. The Authors did not reveal the details of the management of the children and patients with malignancy and PNH. Can they add it to the review ?
Author Response
Reviewer 2
- Can the Authors specify the epidemiology and history of the PNH ?
Reply: We are thankful for this idea. The following phrases were added in the manuscript (2.1 section): “PNH is a rare disorder, with an estimated incidence as high as 15.9 individuals per million worldwide.”
“PNH was first described by European physicians in the latter half of the 19th century, while several observational studies, resulted in the establishment of PNH as a different clinical entity from march hemoglobinuria and paroxysmal cold hemoglobinuria [27].” (Lines 88-92)
- The Authors did not reveal the details of the management of the children and patients with malignancy and PNH. Can they add it to the review ?
Reply: Thanks for this comment. Data regarding the management of PNH in these patients’ groups were incorporated in the manuscript. The following phrases were incorporated:
“In patients with underlying bone marrow failure, the management is similar to those with classical PNH when symptoms of hemolysis are the prominent.” (Lines 403-405)
“PNH is an ultra-rare cause of thrombosis in pediatric patients[177]. In the observational study of Curran et al., 12 consecutive PNH patients were enrolled, and thrombosis developed in 6 of them [178]. TEs occurred in unusual sites, such as portal vein, hepatic veins, and cerebral sinuses. Treatment with C5 complement inhibitors (Eculizumab, Ravulizumab) constitutes also the main option in these patients. In the recently published prospective study of Chonat et al., Ravulizumab was found to be a safe and effective option for children with PNH in both Eculizumab naïve or experienced patients[179]. More real-world studies, examining the role of complement inhibitors in children with PNH are essential. However, this might be a challenging task, given the rarity of PNH in pediatric settings. It has to be highlighted that children at less than 2 years of age, or those without vaccination against N. meningitidis, should get vaccinated before the initiation of treatment and receive prophylactic antibiotics post-vaccination for the appropriate time course [180].” (Lines 724-736)